# Effects of Post-Ruminal Urea Supplementation during the Seasonal Period on Performance and Rumen Microbiome of Rearing Grazing Nellore Cattle

**DOI:** 10.3390/ani12243463

**Published:** 2022-12-08

**Authors:** Mailza Gonçalves de Souza, Irene Alexandre Reis, Isabela Pena Carvalho de Carvalho, Marco Aurélio De Felicio Porcionato, Laura Franco Prados, Yury Tatiana Granja-Salcedo, Gustavo Rezende Siqueira, Flávio Dutra de Resende

**Affiliations:** 1Department of Animal Sciences, São Paulo State University “Júlio de Mesquita Filho” (UNESP), Jaboticabal 14884-900, Brazil; 2Trouw Nutrition R&D, 3811 MH Amersfoort, The Netherlands; 3Trouw Nutrition R&D, Campinas 13080-650, Brazil; 4Agência Paulista de Tecnologia dos Agronegócios (APTA), Colina 14770-000, Brazil; 5Corporación Colombiana de Investigación Agropecuaria (AGROSAVIA), Centro de Investigación El Nus, San Roque, Antioquia 053030, Colombia

**Keywords:** non-protein nitrogen, true protein, recycling of nitrogen, rumen bacteria

## Abstract

**Simple Summary:**

Protein is a crucial nutrient for improving performance and fiber utilization by grazing ruminants. The use of non-protein sources can be used to improve the intake and digestibility of poor-quality forages, since rumen microbes can use NPN to synthesize amino acids. Urea supplementation has become common for cattle. However, urea shows a rapid hydrolysis in the rumen, which can lead to excess ammonia and can impair animal performance and risk intoxication. Supplementation with post-ruminal available urea can prevent ammonia accumulation and can be an alternative source of nitrogen for grazing cattle. The objective of this study was to evaluate the effects of post-ruminal urea supplementation in different types of supplement formulation for Nellore bulls in the growing phase during the seasonal period. Performance and rumen environmental parameters were also evaluated. Our results emphasize that replacing soybean meal as a source of true protein with non-protein nitrogen resulted in reduced performance, and that conventional urea and post-ruminal urea have similar effects on animal performance and rumen metabolism.

**Abstract:**

The objective was to evaluate the effects of urea with post-ruminal absorption in the supplementation of growing Nellore cattle reared on pasture during a seasonal period. For the study, two experiments were conducted. In experiment 1, rumen and blood parameters were evaluated using eight rumen-cannulated Nellore bulls with initial body weight (BW) of 763 ± 44 kg, distributed in a double Latin square 4 × 4. In experiment 2, 120 Nellore steers with initial BW of 380 ± 35 kg were used for performance evaluation, distributed in a randomized block design (blocking factor or initial BW). The evaluated treatments were 1: (TP-U) (control) = supplement with 24% crude protein (CP) containing urea as a source of non-protein nitrogen (NPN; 3%) and soybean meal, 2: (TP-PRU) = 24% CP supplement containing post-ruminal urea (PRU; 3.6%) and soybean meal; 3: (NPN-U-PRU) = 24% CP supplement containing urea + post-ruminal urea (U = 3% and PRU = 3.9%), without soybean meal; 4: (NPN-PRU) = supplement with 24% CP containing post-ruminal urea (7.5%), without soybean meal. The supplement was offered at 3 g/kg BW per animal, daily, once a day. All animals were kept on *Urochloa brizantha* cv. Marandu pasture. Statistical analyses were performed using the SAS PROC MIXED, and the data were evaluated by the following contrasts: C1 = TP-U/TP-PRU vs. NPN-U-PRU/NPN-PRU (Soybean meal replacement by NPN); C2 = TP-U vs. TP-PRU (conventional urea vs. post-immune urea); C3 = NPN-U-PRU vs. NPN-PRU (low and high post-ruminal urea-PRU level). The digestibility of dry matter, organic matter, and NDF was lower when soybean meal was replaced by non-protein nitrogen, also being different between the levels of post-ruminal urea used in the supplement. Ruminal pH was different when soybean meal was replaced by NPN (*p* = 0.003). Total concentration of short-chain fatty acids, concentrations of isobutyrate (*p* = 0.003), valerate (*p* = 0.001), and isovalerate (*p* = 0.001) were different, and blood urea was different when soybean meal was replaced by NPN (*p* = 0.006). Simpson’s diversity index was higher in the rumen of animals supplemented with TP-U than in those supplemented with TP-PRU (*p* = 0.05). A total of 27 phyla, 234 families, and 488 genera were identified. Nitrospirota and Gemmatimonadota phyla were detected just in the rumen of steers supplemented with TP-PRU. The performance (final BW, weight gain and gain per area) of the animals was different, being higher (*p* = 0.04) in animals supplemented with soybean meal, compared to NPN. The removal of soybean meal from the supplement and its replacement with either conventional urea plus post-ruminal urea or only post-ruminal urea compromises the performance of the animals. The lower the post-ruminal urea inclusion level, the lower the apparent digestibility of dry matter, organic matter, and NDF, when compared to animals supplemented with higher levels.

## 1. Introduction

Throughout evolution, animal species developed the ability to conserve nitrogen [1], for ruminants, the symbiosis with microorganisms helped them survive and colonize different parts of the planet while eating diets with low amounts of protein [2]. Another factor that contributed to the evolution of ruminants was the ability to recycle urea and use non-protein nitrogen (NPN) for the synthesis of the high biological values of microbial protein. Thus, the use of alternative sources of nitrogen became a viable option [3,4].

Nitrogen (N) conservation in ruminants occurs through the process of nitrogen recycling in the form of urea from the digestive tract, occurring through saliva and diffusion through the rumen wall [5,6]. In cattle, between 40% and 80% of the urea produced in the liver can return to the gastrointestinal system, with the rumen being the main receptor organ. This recycling is facilitated by rumen microbiota and contributes to the metabolism of ruminal nitrogen. The nitrogen present in the rumen environment is quickly hydrolyzed by the bacterium and NH_3_ is used for the synthesis of the necessary microbes to satisfy the requirements of the ruminants [7,8]. 

Several studies were carried out over time to better understand the nitrogen recycling process, with the displacement of the protein supply, from the rumen to the abomasum, and its real contribution to the metabolism of ruminants, obtaining positive results regarding the improvement in nitrogen use efficiency [9,10,11,12]. These data indicate an advantage in shifting the nitrogen supply from the rumen to the abomasum/intestine, as it stimulates nitrogen recycling and prevents the deleterious effects of ammonia accumulation in the rumen. To our knowledge, this is the first published study that evaluates the performance of animals supplemented with post-ruminal urea.

Urea with post-ruminal release and absorption consists of an encapsulated urea source with reduced rumen release. Approximately 28% of this urea is released in the rumen, with the remainder (72%) released in the post-ruminal compartments. 

To optimize animal performance, it is necessary to formulate supplements that enhance the supply of protein to the animal. One way of enhancing the supply of metabolizable protein in the rumen is to provide non-degraded protein sources [13]. The supply of readily degradable nitrogen sources, such as conventional urea, may not favor the relationship between microbial protein and metabolizable energy, resulting in a voluntary intake reduction due to increased hepatic metabolism as well as blood and intracellular ammonia accumulation, which leads to feelings of discomfort in animals [14].

Considering the above, further studies are needed to understand the use of post-ruminal urea when formulating supplements for grazing cattle. The objective of this study was to evaluate the effects of post-ruminal urea supplementation, testing its use in different supplement formulations for Nellore cattle, during the growing phase in the rainy-dry transition period on animal metabolism and performance.

The hypothesis is that urea utilization can be improved by moving the site of absorption from the rumen to the post-ruminal compartments. Moreover, soybean meal can be completely replaced in the supplement with post-ruminal urea or a combination of post-ruminal urea and conventional urea without compromising intake and performance.

## 2. Materials and Methods

### 2.1. Location

The experiment was conducted at the Alta Mogiana Regional Pole, research unit of Agência Paulista de Tecnologia dos Agronegócios (APTA), in the municipality of Colina, São Paulo, Brazil. The unit is located at 20°42′50″ S and 48°33′52.7″ W. The climate of the region is AW, as per the Köppen classification. The study was carried out between February and May 2021, during the seasonal period between the rainy season and the dry season. During the study, the accumulated precipitation was 108 mm. 

### 2.2. Animals, Area, Experimental Design and Periods

The experiment was conducted according to the guidelines of the National Council for the Control of Animal Experimentation (CONCEA) and was approved by the Ethics Committee in the Use of Animals (CEUA), Faculty of Agrarian and Veterinary Sciences of UNESP—Campus de Jaboticabal (protocol 3974/20). The study was divided into two experiments, one to evaluate rumen parameters (Experiment 1) and another to evaluate animal performance (Experiment 2).

In experiment 1, eight rumen-cannulated male Nellore bulls were used, with an initial body weight of 763 ± 44 kg. The animals were distributed in a 4 × 4 double Latin square. The experiment was carried out in an area of four hectares, divided into four paddocks (one hectare each), containing drinkers and a trough for supplementation. The duration was 84 days, divided into four experimental periods of 21 days, with 14 days of adaptation and seven days of sampling, where on the 15th until 18th day stool sampling was carried out. Urine sampling was carried out on the 18th day. On the 19th day, ruminal fluid sampling was performed. On the 20th day, blood sampling was performed at 0 and 6 h. On the 21st day, blood sampling was performed 3 h after supplementation and sampling was performed with rumen content for microbiology analysis.

In experiment 2, 120 Nellore bulls were used, blocked according to body weight. A randomized block design was used to evaluate the performance of the animals, with an average initial weight of 380 ± 35 kg and an average age of 20 months. The experiment lasted 98 days, with 14 days of adaptation, followed by three periods of 28 days each. The experiment was carried out in an area of 44.7 hectares composed of Urochloa brizantha cv. Marandu and divided into 12 paddocks (3.4–4 hectares each), equipped with drinkers to supply water and a trough to supply supplement. The animals were distributed in 12 paddocks, divided into three blocks, according to the initial weight of the animals in a light one medium one heavy block, with three paddocks per treatment, and 10 animals each, where they were kept in a continuous stocking system, with a variable stocking rate. All animals received the supplement daily at 10 am, offered at 3 g/kg of body weight.

### 2.3. Treatments

Four experimental treatments were used, all supplements being isonitrogenous. Different formulations and nitrogen sources were tested (Table 1). Formulations with conventional urea (U) and post-ruminal urea (PRU) were used, and their combination in the supplement formulations are represented by the following acronyms:(1)TP-U: Indicates the supplement contains true protein (TP), coming from soybean meal, and conventional urea (U);(2)TP-PRU: Indicates the supplement contains true protein (TP), coming from soybean meal, and post-ruminal urea (PRU);(3)NPN-U-PRU: Indicates the supplement contains non-protein nitrogen (NPN), coming from the combination of conventional urea (U) and post-ruminal urea (PRU);(4)NPN-PRU: Indicates the supplement contains non-protein nitrogen (NPN), coming from post-ruminal urea (PRU).

The post-ruminal urea (PRU) product was subjected to an in vitro evaluation to estimate ruminal protection rate. Since the solubility of urea is virtually complete and hydrogenated fat is practically insoluble, weight loss is a simple, robust, and quick method for determining the rumen protection rate of PRU. To simulate rumen stability, 2.5 g of pelletized ruminant feed (without urea and containing 10% PRU) and 5 g of PRU were weighed into Ankom nylon bags and placed into 1000 mL Schott flasks containing 250 mL of McDougall’s buffer solution at pH 6.0 and incubated for 6 h at 39 °C at 100 rpm with an amplitude of 25 mm (horizontal circular motion). The nylon bags were removed from the Schott flask, washed with cold water, and air dried at 39 °C until mass constancy was achieved to determine mass loss. The PRU rumen protection rate (g/kg) was calculated as: 1000 − ((PRU mass loss, g/(initial PRU product mass, g) × urea proportion in initial PRU test product)) × 1000). 

The digestibility was predicted by using the same technique as for ruminal stability, as described above. A two-step process was then used to mimic in vivo abomasal and small intestine incubation, respectively. For the abomasal incubation simulation, the PRU residue from the first step, i.e., rumen stability test, was quantitatively transferred to an empty 1000 mL bulkhead bottle with 250 mL pepsin-containing hydrochloric acid solution preheated to 39 °C and incubated for 2 h at pH 2.0 at 100 rpm with amplitude of 25 mm (horizontal circular movement). After incubation, the contents were filtered through a pleated filter and the residue washed with 20 mL of ice-cold water. For the simulation of small intestine incubation, this residue was then added to 250 mL of prepared pancreatic solution and incubated for 24 h at 39 °C and 100 rpm with an amplitude of 25 mm (horizontal circular movement). The contents of the bottle were filtered through a pleated filter and washed with cold water. The filter was then dried at 39 °C until mass constancy was reached to determine mass loss. The predicted digestibility (g/kg) was then determined as: (PRU mass loss, g)/((initial PRU product mass, g) × (urea proportion in test product)) × 1000. Approximately 28% of this urea is released into the rumen, and the remainder (72%) is released into the post-ruminal compartments. 

### 2.4. Forage Evaluation

The double sampling method was used to determine the forage mass of each experiment using a rising plate meter to correlate the readings with canopy height [15].

Samples were collected from each experimental period using the hand-plucking method [16]. They were then analyzed to determine the nutritive value of the forage. The samples were partially air-dried in at 55 °C for 72 h, then ground in a Willey-type knife mill with 1- and 2-mm mesh sieves.

The analysis of the bromatological values was conducted in the laboratory of the research unit. Dry matter (DM) was analyzed by AOAC’s method 934.01, ether extract (EE) by the Goldfish method (920.39), and crude protein (CP) by the Kjeldahl method (984.13), according to the recommendations of [17]. Neutral detergent fiber (NDF) and acid detergent fiber (ADF) were evaluated, according to [18], with the use of the fiber determiner (TE-149, Tecnal-São Paulo, Brazil). Hence, the conditioned samples were placed in non-woven fabric (TNT) bags, then left for one hour at controlled temperature in contact with the detergent solution. Cellulose was solubilized with 72% sulfuric acid and the lignin content obtained by difference from the ADF [19]. Indigestible NDF was obtained by incubating in the rumen for 288 h (12 days) [20]. The quantitative and qualitative characteristics of the forage were subjected to statistical analysis, and there was no difference between the analyzed treatments (*p* > 0.05), with the averages shown in Table 2.

### 2.5. Intake and Digestibility

Forage and supplement intake were estimated in experiment 1 using chromium oxide (Cr_2_O_3_), titanium dioxide (TiO_2_), and indigestible NDF (iNDF) as markers. The fecal excretion was estimated by placing 10 g/bull/day of Cr_2_O_3_ in the rumen for 10 days and collecting feces (directly from the rectum once daily alternated at 7, 10, 13, and 16 h) during the last four days. Fecal samples were weighed and air dried at 55 °C for 72 h and ground in a Wiley mill to pass through 2- and 1-mm sieves. For each bull, a fecal composite sample was collected in each sampling period. The Cr_2_O_3_ was quantified by atomic absorption spectrophotometer and the fecal excretion was calculated according to [21]. 

The TiO_2_ was added to the supplement at 10 g/bull/day for 10 days, which included 6 days to stabilize the fecal excretion marker and 4 days for sample collection [22]. The fecal samples collected were digested in sulfuric acid, and the curve was prepared with the addition of 0, 2, 4, 6, 8, and 10 mg of TiO_2_, following the reading of the samples in a spectrophotometer according to the methodology described by [23]. For analysis of individual consumption, the following equation was used: Supplement CMS = [g of TiO_2_/g of feces × fecal excretion g/d]/[gTiO_2_/g of supplement]. To estimate the DMI, iNDF was used as an internal marker, being determined after ruminal incubation for 288 h [24]. For analysis, the samples were ground in a 2 mm sieve. DMI was estimated according to the following equation: forage DMI = [fecal excretion g/d × (iMF) − supplement DMI × (iMS)]/[iMH], where iMF, iMS, and iMH are the marker concentrations internal in feces, supplement, and forage, respectively. 

### 2.6. Ruminal Fermentation Parameters

Samples were collected from the dorsal, central, and ventral regions of the rumen on the 19th day of each cannulated animal, and a composite sample was formed. The samples were filtered with two layers of gauze and collected at 0, 3, 6, and 12 h after supplementation to determine pH, ammoniacal nitrogen, and short-chain fatty acids.

Samples for N-NH_3_ analysis were preserved with 1 mL of H_2_SO_4_ (sulfuric acid) and stored at −20 °C until analysis by the phenol-hypochlorite colorimetric method described by [25]. The pH was analyzed shortly after collection, using an electric pH meter (DM-22, Digimed, São Paulo, Brazil). Samples for SCFA analysis (15 mL) were stored at −20 °C, and the concentration of acetate, propionate, butyrate, valerate, isovalerate and isobutyrate were analyzed. The samples were centrifuged at 15,000 g for 15 min at 4 °C, being the supernatant (0.5 mL) and continuing the analyses according to the methodology described by [26]. The calibration curve was performed using the chromatographic standards of acetic acid (99.5%; CAS 64-19-97), propionic acid (99%; CAS 79-09-4), isobutyric acid (99%; CAS 79–31–2), butyric acid (98.7%; CAS 107–92–6), and valeric acid (99%; CAS 109–52–4), without using a pattern. 

### 2.7. Blood Parameters

Blood collection was performed by jugular venipuncture with vacuum tubes without anticoagulant. (BD Vacutainer^®^) on the 20th day at times 0, (before supplementation), and 6 h after supplementation, and on day 21 (3 h after supplementation) of the experimental period. The samples were centrifuged (3080 g for 15 min at 4 °C). Soon after, the serum was returned, taken, and stored in an Eppendorf tube at −20 °C. For the analysis, a composite of the 3 schedules was made. The serum was analyzed for uric acid, creatinine, albumin, total protein, liver enzymes (AST and GGT), and plasma urea nitrogen measured in these samples, using kits from the company (Bioclin^®^) according to the manufacturer’s specifications, (uric acid, code K-139), (urea, code K-056), (albumin, code K-040), (creatinine, code K-222), (total protein, code K-031), (AST, code K-048), and (GGT, code K-060). The readings were performed in an automatic biochemistry analyzer (Sistema de Bioquímica Automático SBA-200; CELM^®^).

### 2.8. Ruminal Bacteria and Archaea Diversity

Samples weighing approximately 50 g per animal (a mix of liquid and solid) were collected through the ruminal cannula on day 21 of each experimental period, 3 h after supplementation, and immediately stored at −80 °C until further analysis.

The bacterial pellet was obtained according to the methodology described by [27]. The metagenomic DNA was extracted from the bacterial pellet using the manufacturer’s instructions for the FastPrep-24 Classic Instrument (MP Biomedicals, Illkirch, France). The DNA yield and quality were evaluated as described by [28]. Duplicate libraries were prepared by PCR amplification of the V3 and V4 regions of the 16S ribosomal RNA gene (16S rRNA) for bacteria using the universal primers 515F and 806R as described by [29]. PCR fragments were purified using the Zymoclean Gel DNA Recovery kit (Zymo Research, Irvine, CA, USA), following the manufacturer’s instructions. The resulting fragments were submitted to sequencing on an Illumina NovaSeq6000 PE 250 platform, resulting in an average of 160,000 readings per sample. Readings were mapped against a 16S rRNA reference database. Sequence trimming was performed by selecting sequences greater than ~470 bp in length with an average quality score greater than 40 based on Phred quality score, and duplicate readings were removed using the Prinseq program [28]. Quantitative Insights into Microbial Ecology (QIIME) version (2022.2.0) software was used to filter readings and determine taxonomic operational units (OTUs) as described by [30]. Significant readings were classified based on the multinomial naive Bayes algorithm to group the OTUs of readings with a cut of 98% and, to assign the taxonomy, the SILVA Ribosomal Database Project (RDP-II) was used.

### 2.9. Urinary Parameters

The purine derivatives in the urine of the animals were used as indicators of microbial protein production [31].

Urine was collected by urination stimulated by urethral massage [32] on the 18th day of each experimental period. Samples were collected 4 h before and 4 h after supplementation according to [33]. Aliquots of 10 mL were collected and later diluted in 40 mL of 0.036 nitrogen solution H_2_SO_4_ for allantoin analysis by the colorimetric method, according [31] and described [34]; creatinine (colorimetric-alkaline picrate method) and uric acid (Trinder enzymatic reaction) readings were performed using commercial kits (Bioclin^®^). The readings were performed in an automatic biochemistry analyzer (Sistema de Bioquímica Automático SBA-200; CELM^®^). 

Urinary creatinine excretion was calculated using the following equation: Daily urinary creatinine excretion (g/day) = 0.0345 × BW^0.9491 [35].

The daily excretion of purine derivatives was calculated as the sum of the concentration of allantoin and uric acid. To calculate this value, the absorbed purine derivatives were used. Absorbed purines and microbial nitrogen flux to the small intestine (NMIC) was calculated according to [36], using the equations: Absorbed purines = (purine derivatives + (0.301 × BW^0.75)/0.80) and microbial N = (70 × absorbed purines)/(0.90 × 0.137 × 1000), where absorbed purines are in mMol/day and NMIC is microbial N flux to the small intestine, in g/day. 

### 2.10. Supplement Disappearance Rate

The disappearance rate of the supplement in the trough was monitored in experiment 2, halfway through each experimental period, every 3, 6, 9, and 24 h after being offered, to evaluate the consumption behavior of the animals in relation to the treatments.

### 2.11. Animal Performance

To determine animal performance in experiment 2, the average daily gain (ADG, kg/day) was determined by the difference between the final fasting body weight of each experimental period and the initial fasting body weight. The animals were weighed at the beginning of the experiment, after fasting for 16 h from solids and liquids, and every 28 days. The gain per area (GA, kg/ha) was evaluated by multiplying the ADG by the number of animals and the number of days in each experimental period, divided by the area of the paddock.

### 2.12. Statistical Analyses

The effects of supplements were tested by orthogonal contrasts, according to the equations. Contrast 1 assessed the use of supplements containing soybean meal, conventional urea, or post-ruminal urea, with supplements where soybean meal was replaced by a conventional urea junction plus post-ruminal urea (−1 −1 1 1) (TP-U/TP-PRU vs. NPN-U-PRU/NPN-PRU), Contrast 2 compared conventional urea-containing supplement with post-ruminal urea supplement (−1 1 0 0), (TP-U vs. TP-PRU). Contrast 3 assessed the urea inclusion levels of post-ruminal supplement (0 0 −1 1) (NPN-U-PRU vs. NPN-PRU).

In experiment 1, the data were analyzed in a Latin square design, and the model included treatments, collection time and their interactions as fixed effect, square effect, period, animal, animal within square as random effects. The ruminal bacteria and archaea abundance data were compared between groups by a Friedman’s Test to analyze TP-U/TP-PRU vs. NPN-U-PRU/NPN-PRU, and a paired Wilcoxon signed rank sum test to compare both TP-U vs. TP-PRU and NPN-U-PRU vs. NPN-PRU. 

Experiment 2 was performed in a randomized block design, with four treatments and three replications (pickets as experimental unit), using initial body weight as blocking criterion, considering treatments as a fixed effect and block as a random effect. Response variables measured more than once in the same experimental unit (body weight, forage supply, among others) were analyzed as repeated measures using the SAS REPEATED. The matrices for each variable were chosen according to the BIC (Bayesian Information Criteria) with its lowest value. All outliers were removed for data analysis.

All data were analyzed using the SAS PROC MIXED (SAS Inst. Inc., Cary, NC, USA), with a previous test of normal distribution (Shapiro–Wilk test) and homoscedasticity of variances (Bartlett’s test). Significance was set at *p* < 0.05 as the critical level of probability for type I error and trend at *p* ≥ 0.05 and *p* ≤ 0.10.

## 3. Results 

### 3.1. Intake and Apparent Digestibility

No differences were perceived for pasture, supplement, dry matter, organic matter, protein, and NDF consumption (Table 3). There was a difference in digestibility between treatments that compared the replacement of soybean meal by non-protein nitrogen. Animals supplemented with soybean meal, when compared to NPN, showed a higher dry matter digestibility (587 g/kg) (*p* = 0.02) and organic matter digestibility (538 g/kg) (*p* = 0.03). As for neutral detergent fiber digestibility (381 g/kg), there was a trend (*p* = 0.09) for these same treatments.

There was a difference in the digestibility of dry matter (*p* = 0.04), organic matter (*p* = 0.04) and NDF (*p* = 0.009), between treatments that compared the level of post-ruminal urea inclusion. The animals supplemented with higher levels of post-ruminal urea (NPN-PRU) showed higher apparent digestibility than animals supplemented with lower levels of post-ruminal urea (NPN-U-PRU).

### 3.2. Ruminal Fermentation

Ruminal pH was higher in steers supplemented with NPN (average pH 6.85) than those supplemented with soybean meal (average pH 6.75), (*p* = 0.003). The lowest pH was observed 12 h after supplementation (*p* = <0.01), the animals supplemented with TP-PRU were the ones that showed the lowest pH value, 12 h after supplementation (6.55), followed by TP-U treatments (6.59), NPN-U-PRU (6.60), and NPN-PRU (6.63).

Time and treatment interacted significantly for ammoniacal nitrogen concentration (*p* = 0.05) (Figure 1). Three and six hours after supplementation for the NPN-PRU and TP-U treatments (*p* = 0.04 three hours; *p* = 0.03 six hours), the animals supplemented with NPN-PRU showed higher concentrations of ammoniacal nitrogen, being 11.83 mg/dL and 10.47 mg/dL, while animals supplemented with TP-U showed values of and 8.36 mg/dL and 7.08 mg/dL respectively.

There was a significant treatment and time interaction between NPN-U-PRU and TP-PRU (*p* = 0.02). Three hours after supplementation, the animals supplemented with TP-U had an average of 12.67 mg/dL, while the animals supplemented with NPN-U-PRU had an average of 8.69 mg/dL. There was also interaction between TP-U and TP-PRU treatments (*p* = 0.01), 3 h after supplementation, with averages of 8.36 mg/dL and 12.64 mg/dL respectively (Figure 1).

For the treatments NPN-PRU and NPN-U-PRU, there was a trend (0.07), the animals supplemented with NPN-PRU had an average of 11.83 mg/dL, and 8.69 mg/dL, for the animals supplemented with NPN-PRU. U-PRU (Figure 1).

For TP-U, TP-PRU and NPN-PRU supplements, the highest concentrations were recorded 3 h after supplementation, with 8.36, 12.67, and 11.83 mg/dL respectively, while NPN-U-PRU showed higher concentrations of NH_3_ 6 h after supplementation (8.96 mg/dL).

The concentration of ammoniacal nitrogen (NH_3_) showed a difference between the supplements containing conventional and post-ruminal urea (*p* = 0.04). Ammoniacal nitrogen was also different between supplements containing different levels of post-ruminal urea (NPN-U-PRU vs. NPN-PRU) (*p* = 0.03) (Table 4).

When soybean meal was replaced with NPN, there was a difference in the concentration of short-chain fatty acids (SCFA; *p* = 0.0006). Higher concentrations of SCFA occurred in treatments that contained soybean meal, with (TP-U/TP-PRU), with an average of 68.65 mMol, while treatments where soybean meal was replaced by NPN (NPN-U-PRU/NPN-PRU), had an average value of 58.6 mMol (Table 4).

When conventional urea was used, a difference in SCFA concentration was found between conventional urea and post-ruminal urea (*p* = 0.04). Lower SCFA concentrations (64.7 mM) were found when conventional urea was used. On the other hand, animals that were supplemented with post-ruminal urea had a mean concentration of 72.6 mMol.

Acetate concentration tended to be higher (*p* = 0.07) for treatments containing NPN, compared to supplements containing soybean meal, also showing a difference in the molar ratio of isobutyrate (*p* = 0.003) valerate (*p* = 0.001), and isovalerate (*p* = 0.001). There was a statistical trend (*p* = 0.09), between treatments with conventional urea and post-ruminal urea. Post-ruminal urea showed a higher concentration of isobutyrate than conventional urea.

Valerate production showed a trend (*p* = 0.05) for treatments with different levels of post-ruminal urea, with higher values when a higher percentage of post-ruminal urea was used. The molar proportion of isovalerate showed interaction between treatment and regimen (*p* = 0.008). At 12 h after supplementation, the animals supplemented with soybean meal had an average of 0.76 mMol/100, while the treatments that contained NPN had an average of 0.61 mMol/100. The acetate propionate ratio showed no difference, for any of the contrasts analyzed. There was no difference in the production of microbial protein between the analyzed contrasts.

### 3.3. Ruminal Microbial Diversity

Animals supplemented with conventional urea (TP-U) had a higher proportion of bacteria and a lower proportion of archaea than animals supplemented with post-ruminal urea (TP-PRU) (Table 5). Simpson’s diversity index was higher in the rumen of animals supplemented with TP-U than in those supplemented with TP-PRU (*p* = 0.05).

A total of 27 phyla (Figure 2), 234 families and 488 genera were identified. Nitrospirota and Gemmatimonadota phyla were detected just in the rumen of steers supplemented with TP-PRU. Both Actinobacteriota and WPS-2 phyla had higher ruminal abundance in steers supplemented with soybean meal containing more supplements than those supplemented with non-protein nitrogen sources (Figure 2b). Thermoplasmatota archaea phylum abundance also tends to increase in soybean meal containing supplements treatments (Figure 2a, *p* = 0.08). Proteobacteria, Myxococcota, and Thermoplasmatota phyla were detected in higher abundance in the rumen of NPN-U-PRU steers than in NPN-PRU. The Spirochaetota phylum levels tends to be higher in the rumen of animals supplemented with NPN-U-PRU than those supplemented with NPN-PRU (*p* = 0.09). Steers supplemented with TP-U tended to have higher ruminal abundance of Elusimicrobiota (*p* = 0.09) and lower abundance of Myxococcota (*p* = 0.06) than steers supplemented with TP-PRU. 

The families Bifidobacteriaceae and F082 had lower abundance in the rumen of steers supplemented with NPN instead of soybean meal, while Bacteroidetes BD2-2, vadin BE97, and Rhodospirillales uncultured families had higher abundance when soybean meal was replaced by NPN. The WPS-2 family was detected only in TP-U (Table 6). Metanomethylophilaceae archaea family tended to increase when NPN supplements were offered (*p* = 0.09).

Comparing conventional urea with post-ruminal urea, both Atopobiaceae and Anaerovoracaceae families had higher ruminal abundance in steers feed TP-U than TP-PRU (*p* = 0.04). In addition, *Bacteroidales RF16_group*, *Bacteroidales uncultured*, *Bacteroidales BS11_gut_group*, *Endomicrobiaceae*, *Acidaminococcaceae*, *Eubacterium coprostanoligenes group*, *WCHB1-41*, and *VadinBE97* tended to have higher ruminal abundance in TP-U steers than TP-PRU. In contrast, *Erysipelotrichaceae* and *Peptococcaceae* families tended to have higher ruminal abundance in TP-PRU than in TP-U.

In the evaluation of the high and low levels of post-ruminal urea supplementation, *Bifidobacteriaceae*, *Corynebacteriaceae*, and *Anaerovoracaceae* families had higher abundance in the rumen of NPN-PRU, while Bacteroidetes *BD2-2*, *Hungateiclostridiaceae*, and *Metanomethylophilaceae* were in higher abundance in the rumen of steers supplemented with NPN-U-PRU. *Myxococcaceae* was detected when NPN-U-PRU was supplemented (*p* = 0.04). The abundance of Atopobiaceae (*p* = 0.09), Christensenellaceae (*p* = 0.07), Clostridiaceae (*p* = 0.07), and Oscillospirales families was higher in the rumen of steers supplemented with NPN-PRU than NPN-U-PRU (*p* = 0.08). In contrast, *Methanomethylophilaceae*, *Syntrophomonadaceae*, *Succinivibrionaceae*, and *Spirochaetaceae* tended to be higher in NPN-U-PRU than NPN-PRU.

At the genera level, the soybean meal replacement by NPN resulted in lower ruminal abundance of genera *Sediminispirochaeta*, *SP3-e08*, *Ruminococcus gauvreauii group*, *WPS-2*, *F082*, *Lachnospiraceae NK3A20 group*, and *Lachnoclostridium*, and higher ruminal abundance of *Bacteroidetes BD2-2*, *Lachnospiraceae_ND3007_group*, *Schwartzia*, *Succinivibrionaceae UCG-002*, *vadinBE97*, *Lachnospiraceae* and *Lachnospiraceae probable genus 10* (Appendix A). A tendency of lower ruminal abundance of the genera *Lachnospiraceae AC2044 group*, *Bifidobacterium*, *Mogibacterium*, *Methanomethylophilaceae uncultured*, *Lactobacillus*, *Roseburia*, *Erysipelothrix*, *Eubacterium cellulosolvens group*, *Absconditabacteriales (SR1)*, *Bifidobacteriaceae uncultured*, *Blautia*, and *Atopobium*, and higher abundance of *Prevotellaceae uncultured*, *Stomatobaculum*, *Eubacterium xylanophilum group*, *PeH15*, *Eubacterium ruminantium group*, and *Prevotellaceae UCG-003* was observed. 

The supplementation with TP-PRU resulted in lower ruminal abundance of the genera Eubacterium nodatum group, Eubacterium hallii group, Mogibacterium, Oscillospira, Intestinimonas, Mailhella, Olsenella and Succiniclasticum, and higher ruminal abundance of Turicibacter was verified when compared with TP-U (Appendix A). TP-PRU supplementation also tended to reduce the ruminal abundance of Corynebacterium, Ruminococcus gauvreauii group, vadinBE97, Bacteroidales BS11 gut group, Stomatobaculum, WCHB1-41, Erysipelothrix, CAG-352, Endomicrobium, Amnipila, Anaerovorax, Eubacterium coprostanoligenes group and Bacteroidales_RF16_group, and the ruminal abundance of Erysipelotrichaceae_UCG-002, and Peptococcaceae uncultured tended to increase. 

When NPN-PRU was compared to NPN-U-PRU, a higher abundance of genera Methanomethylophilaceae uncultured, Saccharofermentans, Bacteroidetes BD2-2, Succinivibrionaceae UCG-002, Eubacterium ruminantium group was observed. Lower abundance of Papillibacter, Mogibacterium, Corynebacterium in NPN-U-PRU (Appendix A) was noted. In addition, Eubacterium, Syntrophomonas, and Treponema genera tended towards a higher ruminal abundance in NPN-U-PRU, while ruminal abuncance in Eubacterium, Clostridium sensu stricto 1, Ruminobacter, Ruminococcus gauvreauii group, Muribaculaceae, Oscillospiraceae uncultured, Christensenellaceae R-7 group, Olsenella, and Amnipilatend tended to be higher in NPN-PRU.

### 3.4. Blood Parameters

Blood urea was lower in treatments without soybean meal (*p* = 0.01), with averages of 31.4 mg/dL for treatments with NPN and 34.95 mg/dL for treatments with soybean meal. For animals supplemented with conventional urea and post-ruminal urea, there was a tendency for urea to be present in the blood (*p* = 0.10). Animals supplemented with conventional urea had higher levels of blood urea than animals supplemented with post-ruminal urea. (Table 7).

The presence of Gamma-glutamyltransferase (GGT) tended (*p* = 0.05) to be lower in animals that received post-ruminal urea in relation to animals that received a supplement containing conventional urea. There was a decreasing movement (*p* = 0.07) in uric acid production when soybean meal was replaced by NPN.

### 3.5. Animal Performance

There was a difference for the average daily gain (kg) between the animals that received supplements that contained soybean meal compared to treatments with NPN (*p* = 0.02). Animals supplemented with soybean meal had a higher average daily gain than animals supplemented with NPN (Table 8). The removal of soybean meal from the supplement, and its replacement either by the combination of conventional and post-ruminal urea, or only by post-ruminal urea, resulted in an average decrease in animal performance of 63 g/day.

For final body weight (kg) there was a difference between treatments which replaced soybean meal by NPN (*p* = 0.04, Table 8). The average final body weight of the treatments with soybean meal was 472.2 kg, while the animals supplemented with NPN had a final body weight of 463.9. The removal of soybean meal from the supplement resulted in a decrease of 8,3 kg in the final body weight of the animals.

The average gain per area also showed a difference between these treatments (replacement of soybean meal by NPN), (*p* = 0.02). Animals supplemented with TP-U and TP-PRU showed a greater gain per area, an average of 69.5 k. Animals supplemented with NPN-U-PRU and NPN-PRU had an average gain per area of 64.5 kg.

The average daily gain showed a period effect (*p* < 0.0001; Figure 3). As the water-drought seasonal period was characterized, the average gain was reduced, which must be attributed to the forage quality, which was also reduced throughout the experimental periods.

### 3.6. Disappearance of the Supplement in the Trough

The time of supplement consumption was different between treatments, showing an effect for treatments that contained soybean meal compared to supplements with NPN (*p* < 0.0001). Animals supplemented with TP-U/TP-PRU consumed the entire supplement up to 6 h after supplementation, unlike animals supplemented with NPN, which took longer to complete consumption. At the end of 24 h, there were no leftovers in any treatment (Figure 4).

## 4. Discussion

Steers supplemented with lower levels of post-ruminal urea (NPN-U-PRU) had lower digestibility of DM, OM, and NDF than those supplemented with higher levels of post-ruminal urea (NPN-PRU). Similar results on NDF digestibility were reported by [11], when feeding urea directly into the abomasum of cattle obtained a higher digestibility for NDF than when urea was supplied in the rumen. Higher forage digestibility was also reported in studies conducted by [9,37], with infusion of urea in the duodenum of sheep. Our results show that the higher level of post-rumen urea favored digestibility, so that these animals had a higher nitrogen supply in the rumen via recycling, and this is supported by the observed increase in the ruminal abundance of *Papillibacter*, *Mogibacterium*, and *Corynebacterium* genera. *Papillibacter* belongs to the rumen epithelium-associated microbial community and could be implicated in rumen homeostasis regulation when hosts pass through a feed restriction [38]. *Mogibacterium* is an important bacterium participating in ammonia assimilation through the rumen epithelial wall [39]. On the other hand, some species of *Corynebacterium* can metabolize urea as a nitrogen source when the ammonia concentration in the medium is limited [40].

When soybean meal (TP-U/TP-PRU) was replaced by NPN (NPN-U-PRU/NPN-PRU), the digestibility of dry matter and organic matter was lower. Thus, a lower concentration of total short-chain fatty acids (SCFA) and branched-chain fatty acids (BCFA) was observed. Soybean meal is a source of true protein of high rumen degradability. It provides branched-chain amino acids that give rise to SCFAs. These fatty acids are essential to potentiate the growth of bacteria that degrade fibrous compounds [41], as these microorganisms depend on ammonia and SCFA for the synthesis of its proteins [42]. This is in line with the higher ruminal abundance of several members of Lachnospiraceae family, as *Ruminococcus gauvreauii group*, *Lachnospiraceae NK3A20 group*, *Lachnoclostridium*, *Eubacterium cellulosolvens group*, and *Blautia*, in steers supplemented with soybean meal instead NPN. Many Lachnospiraceae members are associated with carbohydrate metabolism and cellulolytic activity in cattle [43], and this family could play a key role in the nitrogen retention in beef cattle [42]. In addition, ruminal abundance of Thermoplasmatota archaea phylum and its family Methanomethylophilaceae tends to decrease when true protein as soybean meal was replaced by NPN. It is likely that the lower digestibility and lower SCFA production in the rumen limited the supply of H+ to methylotrophic archaea members that need an external source of H+ to reduce methylated compounds [44].

The Bifidobacteriaceae family also was more abundant in animals supplemented with soybean meal. These bacteria groups are important in the metabolism of complex carbohydrates due to their recognized polysaccharide-degrading capabilities [45]. In addition, the SP3 -e08 genera were found in greater abundance when the animals were supplemented with soybean meal. Hence, these two genera show a positive correlation with the production and transport of short-chain fatty acids [46].

When comparing conventional urea and post-ruminal urea, animals supplemented with post-ruminal urea had a higher concentration of SCFA and lower Simpson’s diversity index in their rumen environment. Simpson’s diversity index varies between 0 and 1, and values close to 0 indicate greater observed diversity [47]. Previous reports have suggested that a higher microbial diversity is associated with the digestion and utilization of cellulose in the rumen [48,49]. Thus, post-ruminal urea supplementation might promote rumen microbiota diversity, increasing the production of SCFA in the rumen. In this sense, Nitrospirota and Gemmatimonadota phyla, was detected just in the rumen of animals supplemented with post-ruminal urea. Gemmatimonadota phylum are predominant in soils and pastures, performing functions in the soil with substrate decomposition and nutrient cycling [50]. On the other hand, Nitrospirota is a metabolically diverse, Gram-negative phylum, including sulfate-reducing and nitrite oxidizer bacteria in anaerobic environments [51]. In addition, a higher ruminal abundance of *Turicibacter*, *Erysipelotrichaceae UCG-002*, and *Peptococcaceae uncultured* bacterial genera was observed in the rumen of steers supplemented with post-ruminal urea. *Turicibacter* is a Gram-positive bacteria commonly detected in the gastrointestinal tracts and feces of ruminants and positively associated with ruminal acetate concentration [52], but its role in the nitrogen metabolism in the host remains unclear. Peptococcaceae is a Gram-positive, anaerobic, coccal family with a known ability to ferment protein decomposition products [53]. 

In contrast, most of detected changes on ruminal abundance at genera level when comparing conventional urea and post-ruminal urea involved the lower abundance of several genera from Fimicutes phylum (e.g., *Eubacterium nodatum group*, *Eubacterium hallii group*, *Mogibacterium*, *Oscillospira*, *Intestinimonas*, and *Succiniclasticum*) in the rumen of steers supplemented with post-ruminal urea. The Firmicutes and Bacteroidota phyla are recognized as the most abundant in the rumen environment of Nellore cattle, since both phyla are more resistant to changes in diets, having greater resistance to acid environments [54]. Bacteroidota are responsible for the digestion of complex carbohydrates, while Firmicutes is the phylum that encompasses several genera of fibrolytic and cellulolytic bacteria [55]. Steers supplemented with conventional urea tended to have higher ruminal abundance of Elusimicrobiota phylum, involving mainly the *Endomicrobium* genus. Previous reports of [44] associated this phylum with the nitrogen metabolism in Nellore cattle, indicating an important role of this group of bacteria in the urinary nitrogen excretion regulation in ruminants. The abundance of rumen archaea was found at 11–15 %. Similar levels have been previously found in pasture-fed animals [56].

Higher pH favors the outflow of ammonia from the rumen into the bloodstream [57]. Thus, steers supplemented with a true protein source had lower pH values than animals supplemented with NPN. Excess urea accelerates the production and absorption of ammonia, and pH plays a direct role in this process, raising ammonia leads to an increase in pH, consequently altering the permeability gradient of the rumen epithelium [42]. Lower pH decreases rumen permeability, which can maintain a more adequate nitrogen supply for microbial growth [6,7]. The lower pH value was observed in animals supplemented with soybean meal, in relation to those containing NPN, which can be attributed to a greater microbial growth and, consequently, a greater production of short chain fatty acids. The Actinobacteria phylum showed higher rumen abundance in steers fed soybean meal, when compared to animals fed NPN. This phylum shows a positive correlation with pH [58].

When evaluating the effect of urea infusion in the abomasum of cattle, the TP-PRU group had a lower pH average. In this line, [12], animals that received urea by direct infusion into the abomasum had lower pH averages, with one positive effect, as previously mentioned, by decreasing the permeability of the rumen and thus maintaining a more adequate nitrogen supply for microbial growth in the rumen. Ref. [11] previously reported higher pH behavior when urea was supplied into the rumen compared to urea infused directly into the abomasum.

Ammonia nitrogen levels were all above 5 mg/dL, recommended by [59], as the minimum level to guarantee adequate microbial activity and growth of cellulolytic bacteria. According to [5], an optimal level of NH_3_ would be 10 mg/100 mL, however NH_3_ values should not be considered fixed since the uptake of ammonia for microbial synthesis depends on the rate of carbohydrate fermentation. Steers supplemented with conventional urea (TP-U) had lower concentrations of ammoniacal nitrogen in the rumen, and higher concentrations of urea present in the blood. Different results were found by [12], who reported that the highest concentration of ammoniacal nitrogen occurred when urea was supplied totally or partially in the rumen. According to [60], there are changes regarding the blood parameters linked to the species, leading to variations between breeds, as well as between the age groups of the animals. These changes would explain the variation of values found in the present work in relation to reference values. Another factor that may have affected blood parameters is the age and weight of the cannulated animals, as they are heavy adult animals.

According to [61], the normal blood urea level of ruminant animals should be between 20 and 30 mg/dL. The concentration of urea in the blood is directly linked to the percentage of protein in the food ingested by the animal. Low values are found in herds that consume diets with low levels of protein, and higher values by those who consume diets with a high percentage of protein, or with an energy deficit [62].

The results of the present work show that changes in forage quality throughout the experimental periods, characterized by the seasonal water to drought transition, had a direct impact on the reduction of animal performance. During the experimental period, the forage underwent changes in its structure, as well as a decline in its nutritional value.

Changes in proportions were also influenced by the presence of animals in the area, since ruminants practice selective grazing with a preference for consuming green leaves, favoring the accumulation of parts of lesser acceptance [63]. The proportion of green leaves reduced along the experimental periods, due to the presence of animals and the water scarcity caused by the decrease in rainfall, which characterizes the transition period.

The average daily gain of the animals was different throughout the experimental periods. With the quality of the pasture diminishing, the weight gain of the animals decreased. According to [64], animals kept in *Urochloa brizantha* pastures show a seasonal pattern, with increasing rates during spring/summer and decreasing during the rest of the year.

Thus, it is emphasized that supplementation must be carried out according to the characteristics presented by the forage throughout the seasons. This is to minimize the impacts that may be caused on the performance of the animals during the phases of water scarcity that culminate in a forage of low quality.

Thus, successful nutritional management must seek to balance the nutritional requirements of animals with the changes undergone by the forage plant throughout the seasonal fluctuations of the year, so that ideal conditions are guaranteed for continuous and uniform animal growth [65]. 

The replacement of soybean meal either by the addition of conventional urea plus post-ruminal urea, or only by post-ruminal urea, resulted in a reduction in the weight gain of the animals. 

Our results emphasize the need for a true protein source to enhance the weight gain of animals supplemented either with conventional urea or post-ruminal urea.

Although there were no supplement refusals throughout the experimental periods, a different intake behavior was observed between supplements containing soybean meal and NPN in experiment 2. It is known that urea limits consumption [66], thus influencing the time of supplement consumption by animals.

Excess urea limits the intake of animals not only because of its sensory characteristics, but also because of physiological reactions, involving metabolic routes, ammonia derived from excess urea in the rumen can pass intact through the liver, and enter the systemic circulation. Therefore, it directly affects organs and tissues, and the concentrations tolerated by the brain are much lower than those supported by the tissues of the portal system. Thus, interference affects brain energy metabolism, the Krebs cycle is inhibited, and the urea cycle is directly dependent on the Krebs cycle, so the excess of circulating ammonia leads to brain malfunction due to energy deficit, causing discomfort to animals [14].

The use of post-ruminal urea as an ingredient for the formulation of supplements for cattle on pasture is a new technology, and positive results were observed when directly infusing nitrogen [11,12]. Therefore, it is necessary to better understand the contribution of post-ruminal urea in animal metabolism and its interaction with other supplement ingredients. 

The hypothesis that urea utilization would be improved by partially changing the site of nitrogen absorption from the rumen to the post-ruminal compartments is supported by the increased SCFA and higher microbial diversity, although animal performance was unchanged.

The complete replacement of soybean meal in the supplement by urea compromised animal performance. 

## 5. Conclusions

The removal of soybean meal from the supplement and its replacement, by adding either conventional urea plus post-ruminal urea or post-ruminal urea alone, compromises animal performance in grazing Nellore cattle during the seasonal period. The replacement of conventional by post-ruminal urea increases microbial diversity but results in similar animal performance and metabolism.

## Figures and Tables

**Figure 1 animals-12-03463-f001:**
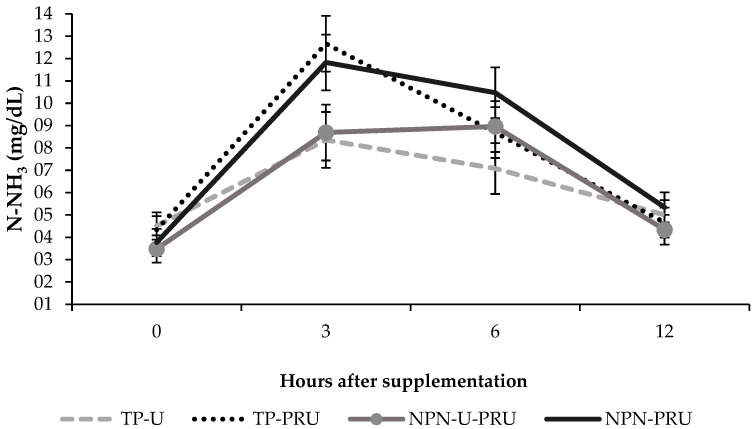
Ammoniacal nitrogen at different times after Nellore cattle receiving different types of supplementation in *Urochloa brizantha* cv. Marandu during the seasonal period.

**Figure 2 animals-12-03463-f002:**
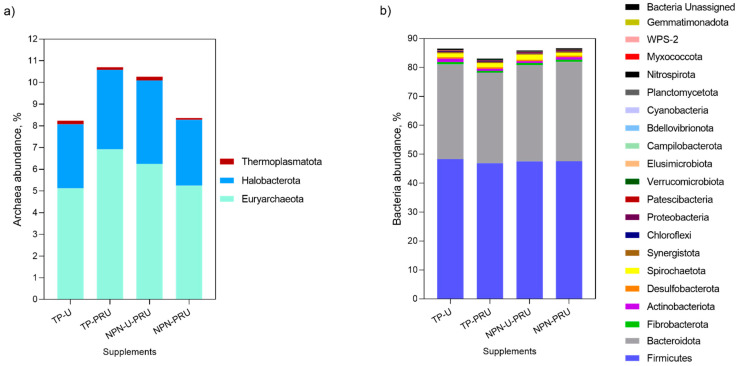
Ruminal archaea (**a**) and bacterial (**b**) abundance at phylum level in grazing Nellore steers, supplemented with post-ruminal urea and conventional urea during the seasonal period (Exp.1).

**Figure 3 animals-12-03463-f003:**
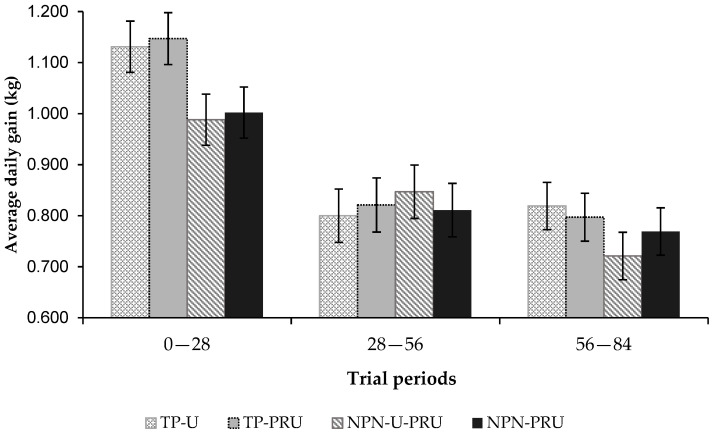
Average daily gain in kg per day, according to the experimental periods. 1° (First)—0 to 28; 2° (Second)—28 to 56; 3° (Third)—56 to 84.

**Figure 4 animals-12-03463-f004:**
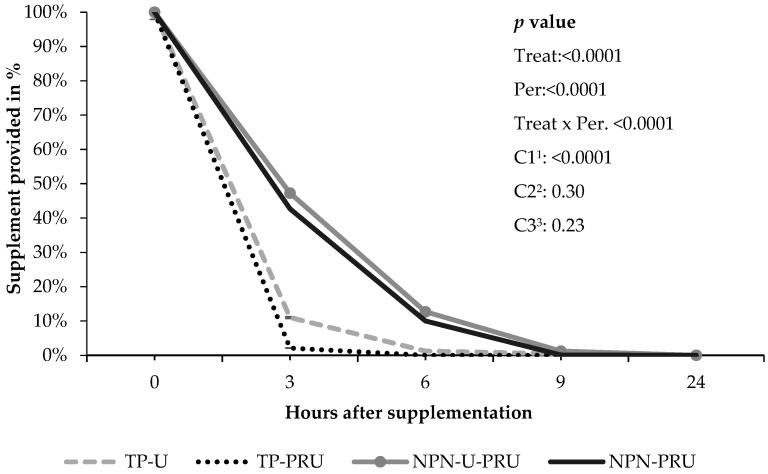
Rate of supplement disappearance over the experimental period of performance animals. Contrasts: ^1^ C1—TP-U/TP-PRU vs. NPN-U-PRU/NPN-PRU (Soybean meal replacement by NPN). ^2^ C2—TP-U vs. TP-PRU (conventional urea vs. PRU); ^3^ C3—NPN-U-PRU vs. NPN-PRU (low and high post-ruminal urea (PRU) level) SEM: 1.21 (error average default.) Hour 0 corresponds to the total amount of supplement provided.

**Table 1 animals-12-03463-t001:** Composition of the supplement used during the experimental period.

	TP-U	TP-PRU	NPN-U-PRU	NPN-PRU
Ingredients (%)				
Soybean meal	24.0	24.0		
Conventional urea	3.0		3.0	
Post-ruminal urea		3.6	3.9	7.5
Corn	61.6	61.0	81.7	81.7
Mineral mix	11.4	11.4	11.4	11.4
chemical composition (g/kg)				
Crude protein	243	249	248	254
Ethereal extract	26.80	30.97	35.53	39.65
Crude fiber	26.79	26.66	17.07	16.95
NPN *	13.80	13.68	28.62	28.50

* Non-protein nitrogen.

**Table 2 animals-12-03463-t002:** Quantitative and qualitative characteristics of *Urochloa brizantha* cv. Marandu during the growing phase of Nellore steers.

	Exp. 1	Exp. 2		
		d 0–28	d 29–56	d 57–84
Quantitative characteristics, whole canopy				
Forage mass (kg DM/ha)	7676	9467	8977	8180
Height (cm)	46.5	46.2	48.1	42.5
Forage supply (kg DM/kg BW)	18.8	8.2	7.8	7.2
Green leaf (%)	20.1	24.3	16.6	7.0
Green stem (%)	16.0	11.3	18.7	22.7
Senescent leaf (%)	48.1	57.8	56.2	27.8
Senescent stem (%)	15.7	6.8	8.8	42.7
Density (kg/m^3^)	2.3	2.3	2.1	1.8
Qualitative characteristics (g/kg DM), hand-plucked samples				
Dry matter	310	287	301	477
Crude protein	90.9	84.3	90.6	61.3
Ether extract	13.9	12.5	11.7	10.8
NDF ^1^	703.2	729.5	729.3	733.9
ADF ^2^	330.8	355.5	337.6	360.8
Lignin	48.4	49.9	52.8	63.5
Indigestible NDF	168.0	157.2	179.8	225.7

^1^ NDF: Neutral detergent fiber; ^2^ ADF: Acid detergent fiber. 1° (First)—0 to 28; 2° (Second)—29 to 56; 3° (Third)—57 to 84.

**Table 3 animals-12-03463-t003:** Intake and digestibility in cannulated grazing Nellore steers (exp.1) supplemented with post-ruminal urea and conventional urea during the seasonal period.

	Treatments				SEM ^1^	*p* Value		
	TP-U	TP-PRU	NPN-U-PRU	NPN-PRU		C1 ^2^	C2 ^3^	C3 ^4^
Total intake (kg/day)								
Forage intake	11.5	10.6	9.8	10.4	0.92	0.23	0.45	0.58
Supplement intake	1.18	1.11	1.05	0.98	0.15	0.35	0.71	0.72
Dry matter	12.7	11.7	10.8	11.4	1.01	0.21	0.45	0.65
Organic matter	11.5	10.8	9.98	10.5	0.93	0.24	0.51	0.65
Protein	1.29	1.24	1.17	1.16	0.11	0.31	0.69	0.95
NDF	8.03	7.47	6.83	7.44	0.65	0.27	0.44	0.44
Digestibility (g/kg)								
Dry matter	585	589	541	576	1.15	0.02	0.83	0.04
Organic matter	533	542	498	530	1.07	0.03	0.59	0.04
Protein	602	617	589	585	2.17	0.30	0.62	0.88
NDF	380	381	348	383	0.89	0.09	0.92	0.009

^1^ SEM: Standard error of the mean. ^2^ C1—TP-U/TP-PRU vs. NPN-U-PRU/NPN-PRU (Soybean meal replacement by NPN). ^3^ C2—TP-U vs. TP-PRU (conventional urea vs. PRU); ^4^ C3—NPN-U-PRU vs. NPN-PRU (low and high post-ruminal urea (PRU) level); NDF: Neutral detergent fiber.

**Table 4 animals-12-03463-t004:** Ruminal fermentation parameters of Nellore cattle receiving different types of supplementations in *Urochloa brizantha* cv. Marandu during the seasonal period (exp.1).

	Treatments				SEM ^1^	*p* Value				
	TP-U	TP-PRU	NPN-U-PRU	NPN-PRU		C1 ^2^	C2 ^3^	C3 ^4^	Hour	Treat × Hour
**pH**	6.79	6.71	6.84	6.86	0.07	0.003	0.12	0.23	<0.01	0.16
N-NH_3_ (mg/dL)	6.24	7.59	6.37	7.85	0.55	0.68	0.04	0.03	<0.01	0.05
VFA Total (mMol)	64.7	72.6	59.5	57.6	6.03	0.0006	0.04	0.62	0.17	0.12
**VFA (mMol/100 mMol)**										
Acetate	75.7	75.5	75.9	76.2	0.50	0.07	0.72	0.34	<0.01	0.32
Propionate	14.6	14.5	14.5	14.2	0.31	0.37	0.80	0.29	<0.01	0.51
Butyrate	7.76	7.97	7.69	7.71	0.25	0.20	0.26	0.90	<0.01	0.18
Isobutyrate	0.65	0.62	0.59	0.60	0.01	0.003	0.09	0.81	<0.01	0.53
Valerate	0.50	0.52	0.43	0.48	0.02	0.001	0.44	0.05	<0.01	0.19
Isovalerate	0.75	0.76	0.69	0.68	0.03	0.001	0.64	0.89	0.004	0.008
A:P relation	5.22	5.22	5.27	5.41	0.14	0.25	0.99	0.35	<0.01	0.52
MP (g/d)	1143	1070	962	720	216	0.23	0.80	0.45	-	-

^1^ SEM: Standard error of the mean. ^2^ C1—TP-U/TP-PRU vs. NPN-U-PRU/NPN-PRU (Soybean meal replacement by NPN). ^3^ C2—TP-U vs. TP-PRU (conventional urea vs. PRU); ^4^ C3—NPN-U-PRU vs. NPN-PRU (low and high post-ruminal urea (PRU) level); VFA volatile fatty acids; N-NH_3_ rumen ammonia; MP microbial protein.

**Table 5 animals-12-03463-t005:** Median and interquartile range of Bacteria, Archaea, in Nellore steers supplemented with post-ruminal urea and conventional urea during the seasonal period (exp.1).

	Treatments				*p* Value		
	TP-U	TP-PRU	NPN-U-PRU	NPN-PRU	C1	C2	C3
Bacteria,%	87.35 ± 4.41	84.72 ± 4.27	85.42 ± 2.48	88.1 ± 4.39	0.53	0.06	0.26
Archaea,%	12.64 ± 4.41	15.27 ± 4.27	14.57 ± 2.48	11.89 ± 4.38	0.53	0.06	0.26
Richness							
ACE	2523.84 ± 122.74	2495.51 ± 248.46	2496.74 ± 254.87	2521.11 ± 180.77	0.82	0.74	0.38
Chao 1	2576.20 ± 104.31	2510.20 ± 230.38	2519.89 ± 265.14	2585.35 ± 188.91	0.74	0.54	0.19
Diversity							
Fisher alpha	496.93 ± 30.55	480.68 ± 42.16	485.33 ± 50.72	496.03 ± 44.86	0.93	0.64	0.19
Shannon	9.91 ± 0.28	9.66 ± 0.22	9.77 ± 0.14	9.90 ± 0.27	0.86	0.23	0.84
Simpson	0.9960± 0.001	0.9950 ± 0.002	0.9955 ± 0.001	0.9965 ± 0.001	0.93	0.05	0.67

C1—TP-U/TP-PRU vs. NPN-U-PRU/NPN-PRU (Soybean meal replacement by NPN). C2—TP-U vs. TP-PRU (conventional urea vs. PRU); C3—NPN-U-PRU vs. NPN-PRU (low and high post-ruminal urea (PRU) level); using a paired Wilcoxon signed rank sum test.

**Table 6 animals-12-03463-t006:** Median and interquartile variation of relative abundance of the family in Nellore steers receiving supplements containing conventional urea and post-ruminal urea during the seasonal period (exp.1).

			Treatments				*p* Value		
Domain	Phylum	Family	TP-U	TP-PRU	NPN-U-PRU	NPN-PRU	C1	C2	C3
Bacteria	Actinobacteriota	*Atopobiaceae*	0.935 ± 0.305	0.587 ± 0.275	0.541 ± 0.142	0.650 ± 0.256	0.20	0.04	0.09
		*Bifidobacteriaceae*	0.166 ± 0.110	0.169 ± 0.127	0.034 ± 0.07	0.137 ± 0.07	0.04	0.88	0.04
		*Corynebacteriaceae*	0.022 ± 0.013	0.010 ± 0.013	0.003 ± 0.011	0.014 ± 0.011	0.22	0.06	0.04
	Bacteroidota	*F082*	4.855 ± 0.922	4.085 ± 1.274	3.873 ± 1.256	3.559 ± 1.207	0.03	0.89	0.48
		*Bacteroidales RF16_group*	0.482 ± 0.046	0.449 ± 0.105	0.484 ± 140	0.435 ± 0.087	0.68	0.09	0.40
		Bacteroidales *uncultured*	0.265 ± 0.080	0.182 ± 0.062	0.219 ± 0.081	0.213 ± 0.077	0.96	0.07	0.33
		*Bacteroidales BS11_gut_group*	0.585 ± 0.185	0.362 ± 0.210	0.511 ± 0.307	0.439 ± 0.202	0.57	0.07	0.48
		*Bacteroidetes BD2-2*	0.124 ± 0.04	0.108 ± 0.03	0.164 ± 0.020	0.127 ± 0.030	0.01	0.44	0.02
	Elusimicrobiota	*Endomicrobiaceae*	0.019 ± 0.012	0.009 ± 0.015	0.022 ± 0.018	0.012 ± 0.010	0.93	0.09	0.23
	Firmicutes	*Christensenellaceae*	11.09 ± 1.88	11.55 ± 1.45	10.85 ± 3.37	12.08 ± 4.13	0.92	0.67	0.07
		*Hungateiclostridiaceae*	1.914 ± 0.399	2.079 ± 0.513	2.418 ± 0.554	2.056 ± 0.571	0.18	0.89	0.04
		*Acidaminococcaceae*	1.690 ± 0.688	1.346 ± 0.189	1.672 ± 0.321	1.634 ± 0.099	0.55	0.09	0.78
		*Eubacterium coprostanoligenes group*	1.343 ± 0.292	1.180 ± 0.404	1.054 ± 0.357	1.174 ± 0.388	0.15	0.09	0.33
		*Anaerovoracaceae*	2.487 ± 0.342	2.056 ± 0.505	1.934 ± 0.380	2.129 ± 0.209	0.20	0.04	0.04
		*Erysipelotrichaceae*	0.105 ± 0.07	0.123 ± 0.329	0.093 ± 0.042	0.092 ± 0.042	0.14	0.07	0.78
		*Clostridiaceae*	0 ± 0.01	0.01 ± 0.148	NI	0.006 ± 0.018	0.17	0.18	0.07
		*Syntrophomonadaceae*	0.0035 ± 0.006	0.002 ± 0.009	0.004 ± 0.006	NI	0.40	0.89	0.07
		*Oscillospirales*	NI	0.007 ± 0.014	NI	0.009 ± 0.011	0.25	0.86	0.08
		*Peptococcaceae*	NI	0.008 ± 0.006	0.005 ± 0.008	0.003 ± 0.007	0.62	0.06	0.46
	Myxococcota	*Myxococcaceae*	NI	NI	0.003 ± 0.007	NI	0.77	0.11	0.04
	Proteobacteria	*Succinivibrionaceae*	0.154 ± 0.128	0.191 ± 0.170	0.181 ± 0.033	0.157 ± 0.024	1.00	0.57	0.09
		*Rickettsiales uncultured*	0.036 ± 0.01	0.050 ± 0.037	0.082 ± 0.058	0.051 ± 0.027	0.12	0.21	0.07
		*Rhodospirillales unculture*	0.011 ± 0.01	0.017 ± 0.013	0.033 ± 0.015	0.022 ± 0.016	0.04	0.35	0.40
	Spirochaetota	*Spirochaetaceae*	1.309 ± 0.31	1.422 ± 0.587	1.751 ± 1.1835	0.987 ± 0.534	0.23	0.48	0.09
	Verrucomicrobiota	*WCHB1-41*	0.026 ± 0.013	0.016 ± 0.020	0.017 ± 0.020	0.023 ± 0.008	0.98	0.08	0.67
		*VadinBE97*	0.013 ± 0.002	0.009 ± 0.004	0.014 ± 0.005	0.015 ± 0.008	0.04	0.07	0.36
	WPS-2	*WPS-2*	0.003 ± 0.012	NI	NI	NI	0.03	0.40	0.32
Archaea	Thermoplasmatota	*Methanomethylophilaceae*	0.157 ± 0.121	0.117 ± 0.153	0.168 ± 0.135	0.081 ± 0.054	0.09	0.67	0.02

C1—TP-U/TP-PRU vs. NPN-U-PRU/NPN-PRU (Soybean meal replacement by NPN). C2—TP-U vs. TP-PRU (conventional urea vs. PRU); C3—NPN-U-PRU vs. NPN-PRU (low and high post-ruminal urea (PRU) level); using a paired Wilcoxon signed rank sum test; NI= Not identified.

**Table 7 animals-12-03463-t007:** Blood parameters of Nellore cattle receiving different types of supplementations in *Urochloa brizantha* cv. Marandu during the seasonal period.

	Treatments				SEM ^1^	*p* Value				
	TP-U	TP-PRU	NPN-U-PRU	NPN-PRU		C1 ^2^	C2 ^3^	C3 ^4^	Hour	Treat × Hour
Urea, mg/dL	36.5	33.4	32.6	30.2	3.30	0.01	0.10	0.20	0.01	0.10
Albumin, g/dL	4.0	3.8	3.7	3.8	0.16	0.22	0.13	0.45	0.41	0.52
Protein, mg/dL	10.2	12.8	9.5	9.8	1.50	0.24	0.22	0.91	0.50	0.51
AST, U/L	98.9	96.2	94.5	93.8	4.45	0.42	0.64	0.91	0.73	0.64
GGT, U/L	21.2	19.5	20.7	20.8	2.49	0.42	0.05	0.84	0.27	0.34
Uric acid, mg/dL	2.6	2.5	2.4	2.4	0.12	0.07	0.45	0.54	0.005	0.14
Glucose, mg/dL	121	112	117	121	6.20	0.53	0.12	0.56	0.73	0.52
Creatinine, mg/dL	2.21	2.18	2.19	2.21	0.05	0.96	0.68	0.71	0.21	0.69

^1^ SEM: Standard error of the mean. ^2^ C1—TP-U/TP-PRU vs. NPN-U-PRU/NPN-PRU (Soybean meal replacement by NPN). ^3^ C2—TP-U vs. TP-PRU (conventional urea vs. PRU); ^4^ C3—NPN-U-PRU vs. NPN-PRU (low and high post-ruminal urea (PRU) level).

**Table 8 animals-12-03463-t008:** Performance of grazing Nellore cattle during the seasonal receiving different types of supplements.

	Treatments				SEM ^1^	*P* Value				
	TP-U	TP-PRU	NPN-U-PRU	NPN-PRU		C1 ^2^	C2^3^	C3 ^4^	Per	Treat × Per
IBW (kg)	395.4	395.5	392.3	391.6	22.62	0.25	0.96	0.86	-	-
FBW (kg)	471.5	472.9	463.9	463.9	22.34	0.04	0.79	0.99	-	-
ADG (kg)	0.916	0.923	0.852	0.861	0.03	0.02	0.87	0.82	<0.01	0.11
CR (UA/ha)	2.49	2.52	2.41	2.45	0.19	0.62	0.88	0.86	<0.01	0.06
GBA (kg/ha)	71	68	62	67	2.77	0.02	0.31	0.14	<0.01	0.14

^1^ SEM: Standard error of the mean. ^2^ C1—TP-U/TP-PRU vs. NPN-U-PRU/NPN-PRU (Soybean meal replacement by NPN). ^3^ C2—TP-U vs. TP-PRU (conventional urea vs. PRU); ^4^ C3—NPN-U-PRU vs. NPN-PRU (low and high post-ruminal urea (PRU) level); IBW= Initial body weight; FBW = Final body weight; ADG = Average daily gain; CR = Capacity rate; GPA= Gain per area.

## Data Availability

Data available on request due to privacy or ethical restrictions.

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
