# Peer review of "Effects of Post-Ruminal Urea Supplementation during the Seasonal Period on Performance and Rumen Microbiome of Rearing Grazing Nellore Cattle"

_animals, 2022, doi:10.3390/ani12243463_

Round 1

Reviewer 1 Report

Dear authors,

the paper has a lot of merits and presents a strong theoretical basis and very relevant and applied results. Below are some suggestions, comments, and questions for a better understanding of the work in question. I separated my review into three sections, one more general (overall comments), one more specific, and one last with errors (mainly typos) that I figured out reading the text and need to be corrected.

Overall comments:

1)      I strongly recommend professional English proofreading. Some phrases are difficult to understand and are not clearly written for the reader's good understanding. As I mentioned, the work has relevant results and deserves better writing so that it is better accepted by the scientific community and reaches more citations.

2)      In addition to professional English proofreading, try to be more explicit about technical jargon or technical descriptions in the materials and methods section. In general, the methods used are not very evident and well described.

Specific comments, suggestions, and/or questions:

1)      The title of the paper is not clear. What does the seasonal period mean? It is difficult for the reader, before reading the paper, to understand well what this means. I would suggest that you think of a more suggestive and clear title in relation to the objectives of the work. For instance, instead of ‘seasonal period’ you may use ‘rainy-dry transition period’.

2)      Line 46 and 47: you mentioned ‘apparent digestibility of NDF’. This is not correct. The digestibility of NDF is the true one, once there is no fiber coming from animal tissues. Please correct this. In table 3, NDF digestibility appears as apparent, I would recommend adding an “*” emphasizing that NDF is the true digestibility.

3)      Line 86 to 88: The origin of this type of urea is unclear. Would it be possible to describe in greater detail how this protection is carried out, or at least offer a bibliographic reference where this same type of urea has been used?

4)      Line 132: how many blocks did you have? What is the reason for adding the initial weight as blocks instead of a covariate in the statistical model?

5)      Line 140: It is not clear what ‘fit animals’ means. Strongly recommend rephrasing all this part (line 140 to 142).

6)      Table 1: Crude protein, ethereal extract, crude fiber, and NPN are not nutrients. You should use a different term for those (i.e., components, etc.).

7)      Line 199: Why didn't you correct the NDF and ADF analyses for ash and protein contaminations?

8)      Line 205: Why did you decide to statistically analyze the qualitative and quantitative characteristics of forages? What type of analysis did you perform?

9)      Line 315: The equation presented is to estimate urinary creatinine excretion, not urine excretion. The BW is powered to 0.9491, however, at least in the pdf version, it looks like a linear coefficient. You should review this. The same issue appears in line 319, where the equation for absorbed purines is presented.

10)   Line 354: add ‘the covariance matrices for each variable …’. What was the criterion for removing outliers? Please describe this in more detail.

11)   Table 3: I would recommend you write the contrasts instead of using C1, C2, and C3. That makes the reading easier.

12)   Figure 3: Why did you use intermediate weightings in the statistical analyses? The 28-day period seems too short to draw conclusions.

Typos and other mistakes:

1)      Line 46: … level);. The apparent…

2)      Line 91: … [18].The …

3)      Line 92: … urea, , may…

4)      Line 122: … aninitial…

5)      Line 155: … coming from…

6)      Line 411: Figure 1. Ammonium nitrogen à ammoniacal nitrogen.

7)      Line 572: Gain by area à gain per area.

8)      Line 579: … periods. . …

9)      Line 748: … Therefore, , it is…

Author Response

Dear reviewer, thanks for the considerations to improve our work, attached follows our return letter.

Author Response

(The authors gave the same response as above.)

Reviewer 3 Report

The paper offers the interesting insights on the effects of urea absorption site in rearing grazing Nellore cattle. It is a good paper. There are some points which need further clarification:

1.      L81 references of 9-17 are too much and too old. Advise to include some up to date references.

2.      The hypothesis of this paper is that urea utilization can be improved by moving the site of absorption from the rumen to the post-ruminal sites. This could be proved by C2 and C3 from statistical analyses in Tables. There are sufficient evidences.

3.      But another objective is “one can completely replace soybean meal in the supplement using post-ruminal urea or a combination of post-ruminal urea and conventional urea” in Introduction. I can not find out any evidence from statistical analyses to (1) compare TP-U and NPN-U-RRU to determine if soybean meal can be replaced by post-ruminal urea under adding conventional urea; (2) compare TP-PRU and NPN-U-RRU to determine if soybean meal can be replaced by conventional urea under adding post-ruminal urea; (3) compare TP-PRU and NPN-RRU to determine if soybean meal can be replaced by post-ruminal urea under adding some post-ruminal urea. (4) OR Control should be all soybean meal without adding any post-ruminal urea and conventional urea and determine whether one can COMPLETELY replace soybean meal using post-ruminal urea or a combination of post-ruminal urea and conventional urea. So, I think to either add new comparison evidence from statistical analyses or delete this objective in Introduction.

4.      I cannot find out details about the products of post-ruminal urea and conventional urea. Add the information about the company, ingredients and composition of two products to explain why EE increases in NPN-U-PRU and NPN-PRU in Table 1.

5.      Table 1: add the ME or NEg because of changes in starch and fat.

6.      Some errors:

L75 N replace Nitrogen

L92 two comma

L155 coming from

L747 directly infusing

L761 increases but results

Author Response

(The authors gave the same response as above.)

Reviewer 4 Report

The manuscript investigated the effects of post-ruminal urea supplementation during the seasonal period on performance and rumen microbiome of rearing grazing Nellore cattle. However, some modifications are needed.

1. The effects of different treatments on rumen microbiome are discussed in detail, but there are no results related to rumen microbiome in the abstract.

2. This description The bromatological analysis were carried out in the laboratory of the research unit where the contents of dry matter (DM) method 934.01, ether extract (EE) by the GoldFish method (920.39) and crude protein (CP) by the Kjeldahl method (984.13) were determined. was incorrect.

3. The abbreviation of acid detection fiber in line 199 is FDA. Why is the abbreviation of acid detection fiber ADF in Table 2?

4. In Table 6, NI should be clearly annotated.

5. On lines 483 to 489, compared with TP-PRU group, there is no significant difference in the relative abundance of Bacteroidales RF16_group, Bacteroidales uncultured, Bacteroidales BS11_gut_group, Endomicrobiaceae, Acidaminococcaceae, Eubacterium coprostanoligenes group, WCHB1-41, VadinBE97 in TP-U group (P value>0.05), so P value should be clearly marked.

6. The references should be in uniform format.

Author Response

(The authors gave the same response as above.)

Reviewer 5 Report

The paper is very interesting with several parameters and different aspects evaluated.  I appreciate how the authors depicted and discussed their results which in my opinion could represent a strong background for future researches.  I only fund a little bit difficult to read some points due to the similarity of the acronyms choosen for the treatments. For istance, I think the authors did a mistake at line 386 describing the ph values 6.59 which is in TPU and not TPRU....U. Thus I just suggets to verify this marginal aspect.   

Author Response

(The authors gave the same response as above.)

Round 2

Reviewer 2 Report

The manuscript has been greatly improved after careful revision by the author. The experimental data is substantial and the method is elaborated clearly, which makes it easier for readers to understand the content. The author also responded well to the questions raised by the reviewers, and some errors in the manuscript have been well corrected.